# A globally sampled high-resolution hand-labeled validation dataset for evaluating surface water extent maps

Rohit Mukherjee[1, *], Frederick Policelli[2, *], Ruixue Wang[3], Elise Arellano-Thompson[3], Beth Tellman[3], Prashanti Sharma[3], Zhijie Zhang[3], and Jonathan Giezendanner[4]

[1]Pacific Northwest National Laboratory, Richland, WA, United States of America
[2]University of Arizona, Tucson, Arizona, United States of America
[3]NASA Goddard Space Flight Center, Maryland, United States of America
[3]Massachusetts Institute of Technology, Boston, Massachusetts, United States of America
[*]These authors contributed equally to this work.

**Correspondence:** Rohit Mukherjee (rohitmukherjee@live.com)

**Abstract.** Effective monitoring of global water resources is increasingly critical due to climate change and population growth. Advancements in remote sensing technology, specifically in spatial, spectral, and temporal resolutions, are revolutionizing water resource monitoring, leading to more frequent and high-quality surface water extent maps using various techniques such as traditional image processing and machine learning algorithms. However, satellite imagery datasets contain trade-offs that result in inconsistencies in performance, such as disparities in measurement principles between optical (e.g. Sentinel-2) and radar (e.g. Sentinel-1) sensors, and differences in spatial and spectral resolutions among optical sensors. Therefore, developing accurate and robust surface water mapping solutions requires independent validations from multiple datasets to identify potential biases within the imagery and algorithms. However, high-quality validation datasets are expensive to build, and few contain information on water resources. For this purpose, we introduce a globally sampled, high spatial resolution dataset labeled using 3-meter PlanetScope imagery. Our surface water extent dataset comprises 100 images, each with a size of 1024x1024 pixels, which were sampled using a stratified random sampling strategy covering all 14 biomes. We highlighted urban and rural regions, lakes, and rivers, including braided rivers and coastal regions. We evaluated two surface water extent mapping methods using our dataset - Dynamic World (Brown et al., 2022) based on Sentinel-2, and the NASA IMPACT model (Paul and Ganju, 2021) based on Sentinel-1. Dynamic World achieved a mean IoU of 72.16% and F1 score of 79.70%, while the NASA IMPACT model had a mean Intersection over Union (IoU) of 57.61% and F1 score of 65.79%. Performance varied substantially across biomes, highlighting the importance of evaluating models on diverse landscapes to assess their generalizability and robustness. Our dataset can be used to analyze satellite products and methods, providing insights into their advantages and drawbacks. Our dataset offers a unique tool for analyzing satellite products, aiding in the development of more accurate and robust surface water monitoring solutions.

# 1 Introduction

Mapping surface water is becoming increasingly important due to the impacts of climate change, as many regions face the prospect of droughts (Dai, 2013) and floods (Tellman et al., 2021). Timely, accurate, and reliable monitoring of surface water extent is critical for better management, conservation, and risk reduction practices, but remains a growing challenge for researchers. Remotely sensed satellite data have provided a unique vantage point for measuring surface water extent (Bijeesh and Narasimhamurthy, 2020; Mueller et al., 2016) using different measurement principles such as optical and radar sensors (Markert et al., 2018). Recent advances in satellite sensors have increased spatial, spectral, and temporal resolutions, leading to significant growth in methods for monitoring surface water using multiple satellite products (Pekel et al., 2016; Martinis et al., 2022; Giezendanner et al., 2023). Among these methods, machine learning and deep learning algorithms gained popularity due to their ability to leverage large volumes of satellite data (both public and commercial) to accurately map the Earth's surface (Isikdogan et al., 2017; Wieland et al., 2023).

However, the effectiveness of satellite water products based on different sensors is not consistent across all conditions, as each product involves trade-offs between spatial, spectral, and temporal resolutions (Wulder et al., 2015). Higher spatial resolution products like PlanetScope (PS) often produce more accurate maps than lower resolution Sentinel-2 (10 m) or Landsat 8 (30 m) (Acharki, 2022). Moreover, radar and optical sensors measure surface water properties differently, leading to variations in accuracy and suitability (Martinis et al., 2022) even at similar spatial resolutions. The study by Ghayour et al. (2021) compared Landsat 8 and Sentinel-2 and found performance varied across methods. As Wolpert (2002) asserted, no single algorithm is expected to perform optimally in every situation. The study by Li et al. (2022) summarizes the current common methods of water extraction based on optical and radar images.

Independently evaluating satellite products and methods using independent validation datasets is crucial for increasing trust in the results (Bamber and Bindschadler, 1997). However, such datasets are resource-intensive to create and existing ones may not be suitable for all needs. For example, BigEarthNet (Sumbul et al., 2019) contains around 600,000 multi-labeled Sentinel-2 image patches, of which 83,000 contain water bodies. This dataset confirms the presence of water within a patch but does not delineate it at the pixel level. The Chesapeake Conservancy Land Cover dataset (Chesapeake Bay Program, 2023) provides high-resolution (1 m) per-pixel water labels for the Chesapeake Bay watershed regional area. LandCoverNet (Alemohammad and Booth, 2020) contains global 10-meter resolution data from Sentinel-2 with a water class. Flood mapping has also been a strong research focus, with datasets like the Sentinel-1-based NASA Flood Detection (Gahlot et al., 2021), Sen1Floods11 (Bonafilia et al., 2020), Sen12-Flood (Rambour et al., 2020), and C2S-MS Floods (Cloud to Street et al., 2022) that use both optical (Sentinel-2) and radar (Sentinel-1) imagery. While suitable for validating surface water maps, some of these datasets rely on 10-meter resolution public satellite imagery or lack global coverage at high resolution. The ephemeral nature of floods also requires specialized detection models even though floodwater is technically surface water (Bonafilia et al., 2020). Wieland et al. (2023) developed a semi-automated global binary surface water reference dataset with 15,000 tiles (256 × 256 pixels) sampled from high-resolution (≤1 m) imagery. However, this dataset uses weak labels generated by a model rather than manual labeling, making it less suitable for validation.

To thoroughly evaluate a product's effectiveness and robustness, multiple independent assessments are needed since high accuracy on one dataset does not guarantee similar performance on others. No single dataset can fully represent the real world (Paullada et al., 2021) and manual labels inevitably contain some subjectivity (Misra et al., 2016). Independent evaluations also help mitigate the issue of data leakage, where the validation set is improperly used during model training, leading to overfitting (Vandewiele et al., 2021). Multiple independent validation datasets are therefore essential for comprehensively evaluating and building trust in remote sensing-based surface water products and methods.

In this study, we present a high-quality, globally sampled, high-resolution surface water dataset consisting of 100 hand-labeled 1024×1024 pixel PlanetScope images at 3-meter resolution. Our work builds upon existing satellite-based datasets for validating surface water extent. The motivation is to provide a higher resolution hand-labeled dataset for evaluating surface water products derived from medium-resolution public satellites like Landsat and Sentinel and commercial higher resolution Planet imagery. Our dataset addresses some of the limitations of existing datasets by providing pixel-level water hand labels at a higher resolution (3 meters) compared to some other datasets and encompassing diverse biomes and contexts (urban/rural, mountains/plains, rivers/lakes) for comprehensive evaluations. We evaluate two state-of-the-art surface water extent mapping methods using our dataset: the Dynamic World land use and land cover product based on optical Sentinel-2 imagery and the NASA IMPACT inundation mapping model based on radar Sentinel-1 data, which was the winning solution in a recent flood detection challenge. By applying our validation dataset to these products and methods, we aim to better understand their advantages and limitations. We anticipate our dataset will contribute to improved accuracy assessment, spatial generalizability analysis, and robustness evaluation of existing surface water products and methods. These advancements can ultimately benefit by promoting more effective monitoring and management of water resources, especially in the face of climate change and population growth.

## 2 Data Preparation

### 2.1 Sampling

Our objective was to build a dataset that closely represents the true distribution of surface water features using only 100 samples. A representative dataset enables testing the spatial generalizability and accuracy of surface water extent products. However, achieving a true representation is nearly impossible (Paullada et al., 2021). We approached this challenge by sampling from different biomes, as defined by Olson et al. (2001), which encompass various climates and land conditions, giving a better chance of providing high variance within samples.

We employed a stratified random sampling strategy to ensure the representativeness of our dataset. First, we created a 2 km buffer around global rivers and lakes shapefiles provided by World Wildlife Fund (2005) using Quantum GIS (QGIS). We then clipped these buffers with the shapefiles of each of the 14 biomes. Within each biome, we randomly placed 50 points using QGIS's random point generator and selected at least 5 of them as samples.

To address the various contexts in which surface water exists, we randomly selected additional samples from urbanized regions (Patterson and Kelso, 2012), braided rivers, and coastal regions. Urban areas are spatially heterogeneous, often re-

sulting in increased complexity for water detection. We also separately sampled from lakes and rivers to ensure a balanced representation of both water body types. Braided rivers and coastal areas were included.

Figure 1 shows the number of samples for each biome, while Figure 2 illustrates the global spatial distribution of the samples. The number of samples from Tropical & Subtropical Dry Broadleaf Forests and Tropical & Subtropical Coniferous Forests was limited due to their smaller area coverage. Approximately two-thirds of our labels are from rivers, and the remaining one-third are from lakes. We sampled a larger portion from Deserts and Xeric Shrublands (16 samples) because water extraction methods generally perform worse in these regions, especially when using radar imagery (Martinis, 2017).

The temporal distribution of our samples spans from 2021 to 2023, covering different seasons to capture seasonal variations in surface water extent. While our sampling strategy aimed to maximize representativeness within the constraints of labeling resources, we acknowledge that the limited number of samples (100) may not fully capture all global surface water variations.

During the sampling process, we implemented quality control measures to ensure that the selected locations were suitable for labeling and analysis. We downloaded the Planet scene for each location, divided the scene into 1024×1024 sized images, and then selected the image that contained sufficient water and no cloud cover.

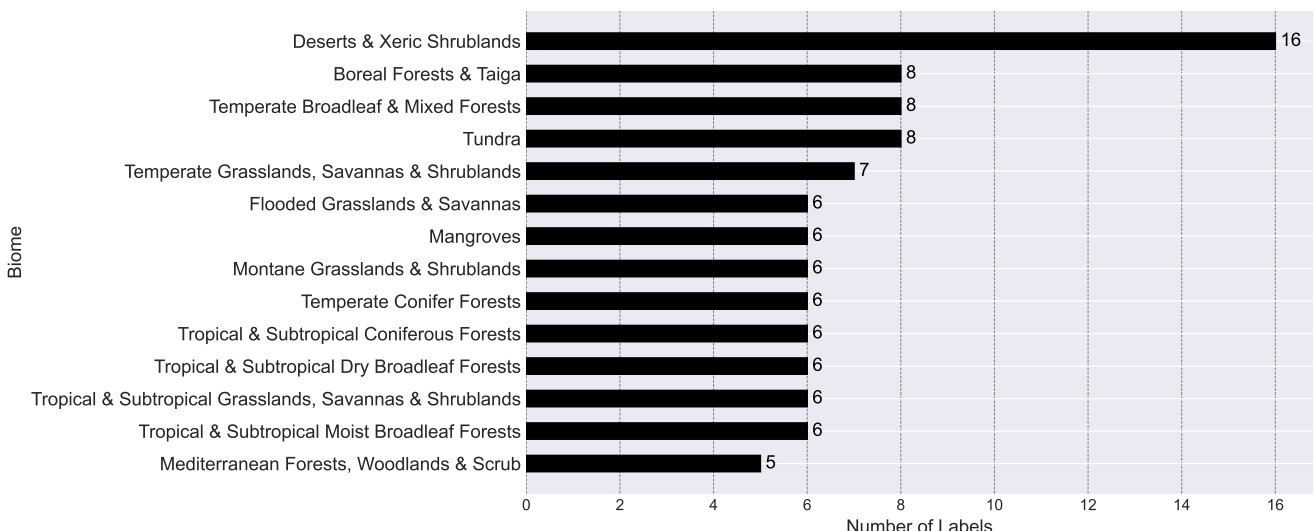

**Figure 1.** Distribution of sampled labels across different biomes. The bar chart illustrates the number of surface water labels collected from each of the 14 biomes defined by Olson et al. (2001). The sampling strategy aimed to ensure a balanced representation of surface water features across diverse ecological regions while accounting for the areal coverage of each biome.

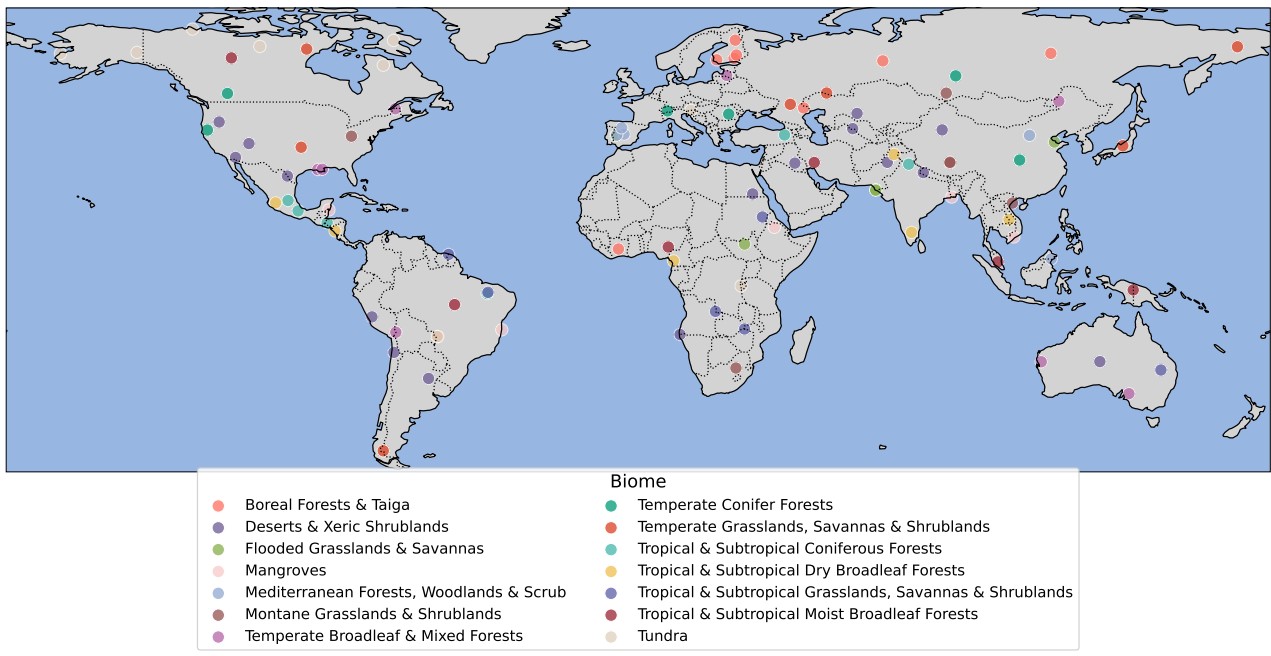

**Figure 2.** Global distribution of the 100 surface water labels sampled for the dataset. The map depicts the geographical locations of the sampled labels, which were sampled to represent diverse global biomes (refer to Table 1 for the number of labels per biome) and ensure a representative dataset of water features. The sampling approach also aimed to capture the variability of surface water features across urban areas, braided rivers, and coastal regions.

## 2.2 Data Processing

After selecting 100 locations based on our sampling strategy, we downloaded 8-band, 3-meter resolution SuperDove PlanetScope (PS) imagery from 2021 to 2023 using our access to the NASA Commercial Smallsat Data (CSDA) Program. As our objective was to evaluate most medium-resolution satellite sensors, including Sentinel-1 (S1), we ensured that the failure of the Sentinel-1B satellite on December 23, 2021, did not create a large temporal gap between the label and the last available scene from the satellite. For locations only covered by Sentinel-1B and not Sentinel-1A, we acquired PS scenes before the Sentinel-1B failure date.

During the scene selection process, we excluded areas with perennially frozen water. If a location contained seasonal ice, we replaced that PS image with a summer image when the water was not frozen. This approach ensured that our dataset focused on liquid water surfaces, which are more relevant for surface water extent mapping.

From each larger PS scene, we extracted a 1024x1024 pixel image, covering an area of approximately 9.4 square kilometers. We chose 1024x1024 pixel images to ensure sufficient pixels and spatial context for comparison with medium-resolution imagery (e.g., Landsat, Sentinel). For instance, a 30-meter Landsat image corresponding to our labels would have around

100x100 pixels, while a 10-meter Sentinel image would have approximately 376x376 pixels. Figure 3 showcases two examples of the PS images selected for labeling, displayed in False Color Composite (near-infrared, red, and green bands).

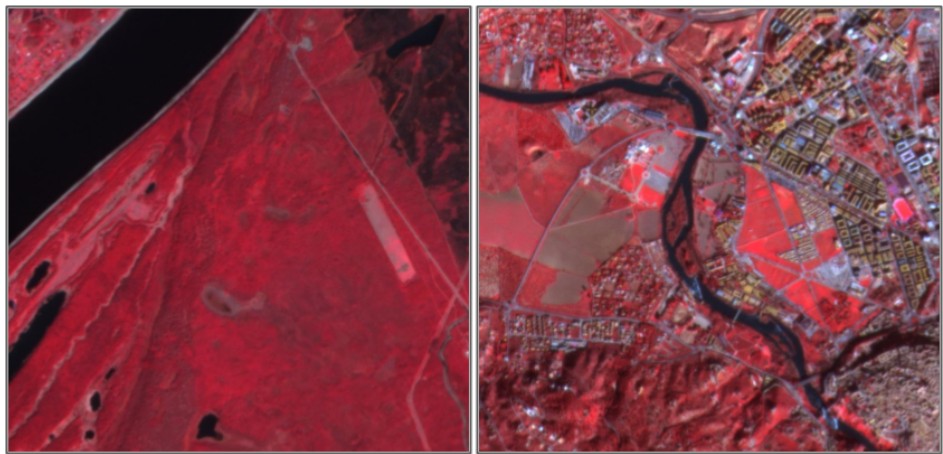

**Figure 3.** PlanetScope images selected for labeling are shown in False Color Composite (near infrared, red, and green). Left: Vilyuy River, Sakha Republic, Russia (SID09) and Right: Tagus River, Toledo, Spain (SID17).

## 2.3 Data Labeling

We used high-resolution 3-meter PlanetScope (PS) data for labeling, ideal for the evaluation of lower-resolution satellite products such as Sentinel-1 (S1), Sentinel-2 (S2) at 10 meters, or Landsat sensors at 30 meters.

The labeling was performed by experienced analysts to distinguish between three classes: water, low-confidence water, and non-water. The water class represents areas with a clear presence of water, while the low-confidence water class marks pixels where the presence of water is uncertain but probable. The non-water class encompasses all other land cover types. To assist the labelers, we provided true-color composite (TCC) and false-color composite (FCC) images using the near-infrared, red, and green bands, for each sample.

In cases where the presence of water was unclear in the PS imagery, we cross-referenced them with higher-resolution basemaps from Bing and Google. Unresolved features were assigned to the low-confidence water category, ensuring that the water class only includes pixels with a high degree of certainty. During the evaluation process, the low-confidence water class can be excluded or added to the water category as necessary.

To streamline the labeling process and ensure the creation of high-quality labels, we utilized the Labelbox platform (Sharma et al., 2019), which provides efficient tools for data annotation. After the initial labeling, we performed several rounds of quality checks on each label to maintain accuracy and consistency across the dataset.

In total, we labeled 100 images, each with a size of 1024×1024 pixels, covering a total surface area of 940 square kilometers. The labeling process, including quality control, took approximately 2 hours per image, resulting in a total of 200 hours of work.

The labeled surface water accounts for nearly 250 square kilometers of the total area. Each label is assigned a unique sample ID (SID) ranging from 1 to 100 and includes the date (YYYYMMDD) of the PS image used for labeling.

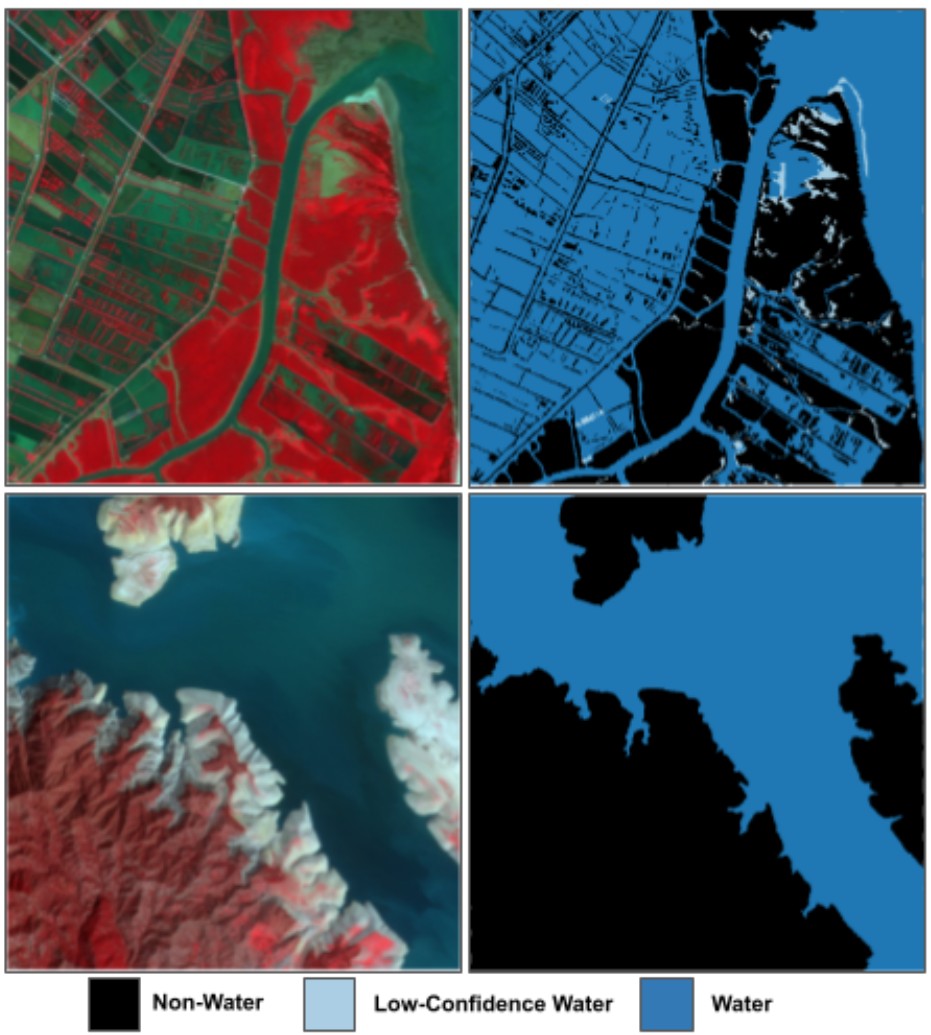

**Figure 4.** Examples of PlanetScope imagery and corresponding labels (Top Row: Dong Tranh River, Ho Chi Min City, Vietnam (SID46), and Bottom Row: Siran River, Pakistan (SID28)). The images are labeled with three categories: 1) non-water, 2) low-confidence water, and 3) water. The low-confidence water category marks pixels where delineating between water and no water is not apparent, but the probability of water being present is moderately high.

## 2.4 Dataset Analysis

We labeled a total of 100 1024x1024 PS images at 3-meters, with the overall class distribution showing that covers 24.9% of the total surface area, low-confidence water covers 1.2%, and the rest (73.9%) is non-water (Fig 5). The distribution of

water pixel percentages for each individual label, as displayed in Figure 6, demonstrates that most labels contain less than 50% water pixels by design, with the mean water surface area per label being 26.10 km$^2$. This focus on having more non-water area enables better delineation of water boundaries, as the water class itself tends to be more homogeneous and therefore less complex from both labeling and mapping perspectives.

As mentioned previously, our labeled dataset covers water surface areas across different biomes (Table 1). The mean percentage of water content per label varies substantially between biomes, from a low of 5.29% for Mediterranean Forests, Woodlands & Scrub to a high of 42.95% for Temperate Grasslands, Savannas & Shrublands. This demonstrates the diversity of landscapes and water coverage captured in our dataset. In total, our dataset provides 2609.78 km$^2$ of labeled water surface area, covering a variety of landscapes such as rivers passing through urban regions, braided rivers in deltas, rivers passing through forests and agricultural fields, and waterbodies in plain and mountainous regions. The diversity and representativeness of our dataset make it a valuable resource for testing the limits and robustness of satellite data products and mapping methods.

## 2.5 Dataset Structure

All 100 labels are in the GeoTIFF format with the UInt8 data type and a single band. Each pixel can contain 4 possible values: 0 (nodata), 1 (non-water), 2 (low-confidence water), and 3 (water). The labels are in the WGS84 (EPSG:4326) coordinate reference system. Each label has a corresponding PlanetScope image used for labeling in Labelbox. The PlanetScope images are also in the WGS84 (EPSG:4326) CRS and contain three spectral bands (red, green, and blue) in true color composite. Based on our PS image release agreement with Planet, we converted the original surface reflectance values to byte format with possible pixel values between 0 and 255, instead of UInt16.

The label files are named using the following convention: 'SIDX_YYYYMMDD.tif', where 'SIDX' is the unique sample ID (X ranging from 1 to 100) and 'YYYYMMDD' represents the date of the PlanetScope image used for labeling. The corresponding PlanetScope images follow the naming convention: 'SIDX_PSID.tif', where SIDX is the same as the label, but PSID is the original SuperDove PlanetScope image ID, allowing for the retrieval of the original surface reflectance values, provided there is access.

Our dataset is organized using the Spatio-temporal Asset Catalog (STAC) format, which is a standardized way to describe and catalog geospatial data. The STAC format provides a clear and consistent structure for storing and accessing the labels and their corresponding PlanetScope images, along with relevant metadata.

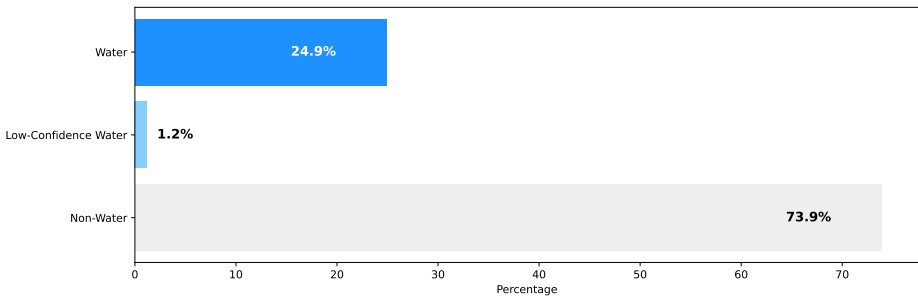

**Figure 5.** Class distribution across Labels (non-water, low-confidence water, and water) for all chips. Non-water class shares the largest percentage as it encompasses the water class. Low-confidence water pixels are only a minor percentage.

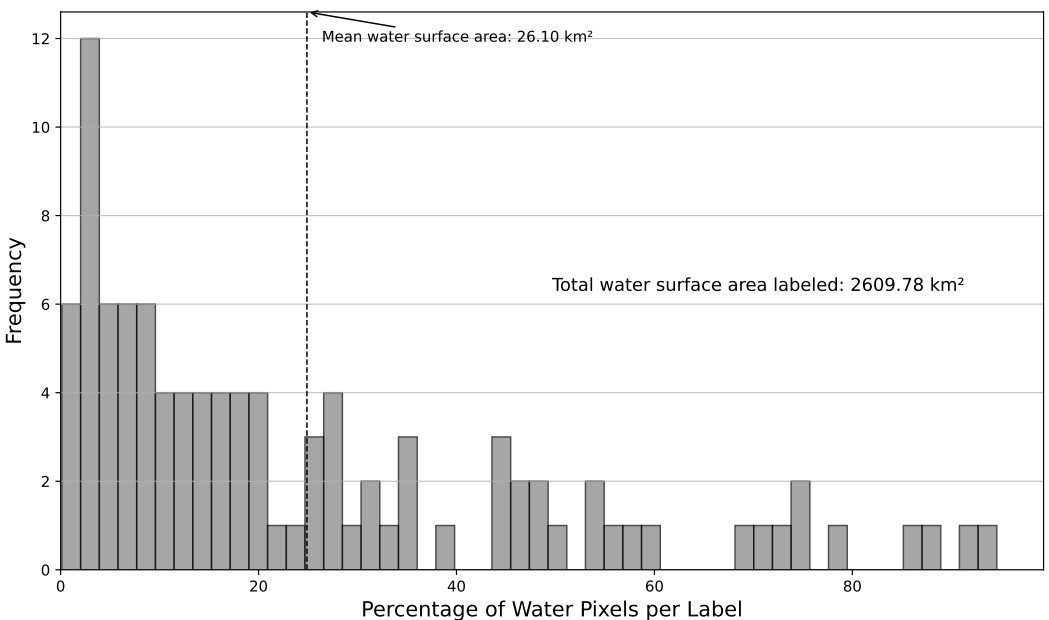

**Figure 6.** Distribution of water pixels per sample. The figure shows the percentage of water pixels within one sample. Most samples contain less than 50% of water by design, as the focus is to delineate the boundaries since the water class is more homogeneous, therefore, less complex.

## 3 Evaluating surface water mapping methods using our hand-labeled dataset

We evaluated two surface water mapping methods based on an optical and a radar satellite imagery product to demonstrate the use of our validation dataset. We used standard metrics for classification - Precision, Sensitivity, Specificity, F1, IoU, and

| Biome | Mean Water Content per Label % |
|---|---|
| Boreal Forests & Taiga | 22.48 |
| Deserts & Xeric Shrublands | 18.96 |
| Flooded Grasslands & Savannas | 27.45 |
| Mangroves | 40.75 |
| Mediterranean Forests, Woodlands & Scrub | 5.29 |
| Montane Grasslands & Shrublands | 23.71 |
| Temperate Broadleaf & Mixed Forests | 19.48 |
| Temperate Conifer Forests | 6.55 |
| Temperate Grasslands, Savannas & Shrublands | 42.95 |
| Tropical & Subtropical Coniferous Forests | 16.80 |
| Tropical & Subtropical Dry Broadleaf Forests | 20.71 |
| Tropical & Subtropical Grasslands, Savannas & Shrublands | 11.96 |
| Tropical & Subtropical Moist Broadleaf Forests | 27.39 |
| Tundra | 30.77 |

**Table 1.** Mean percentage of water content per label across different biomes. The table shows the average proportion of water pixels within the labeled samples for each biome, highlighting the variability in water coverage across diverse ecological regions.

Accuracy for evaluating the two surface water maps. We measured their performance across each biome and their overall performance.

### 3.1 Performance of Sentinel-2 based Dynamic World on detecting surface water

Dynamic World (DW) is a land use land cover product from Google that utilizes a deep learning model trained on their own labeled dataset. The product includes 9 classes, including water, and produces a map for every Sentinel-2 image. Each Sentinel-2 image is post-processed and cloud-removed. We downloaded Sentinel-2 images within 3 days of each of the 100 labeled PlanetScope images. We also applied a Not-a-Number (NaN) filter, ensuring that images with at least 90% valid pixels are considered. After applying the temporal and NaN filters, there were 53 corresponding Sentinel-2 based DW maps out of our 100 labels. From each DW map, we extracted the first band, which contains the water class. Each DW class contains continuous values between 0 and 1, where 1 denotes the highest confidence in the model prediction. We converted the continuous values to binary, thresholding at 0.3. The water class is one of the least confused classes in the DW product, so mixed pixels are less likely. Finally, we evaluated DW on our labels. Note that for evaluation, we converted the low-confidence water class to water. We finally resampled the DW water class to match the resolution of the labels at 3-meters using nearest neighbor interpolation before evaluating. Note that for evaluation, we merged the low-confidence water class with water. Therefore, labels were either 0 (non-water) or 1 (water).

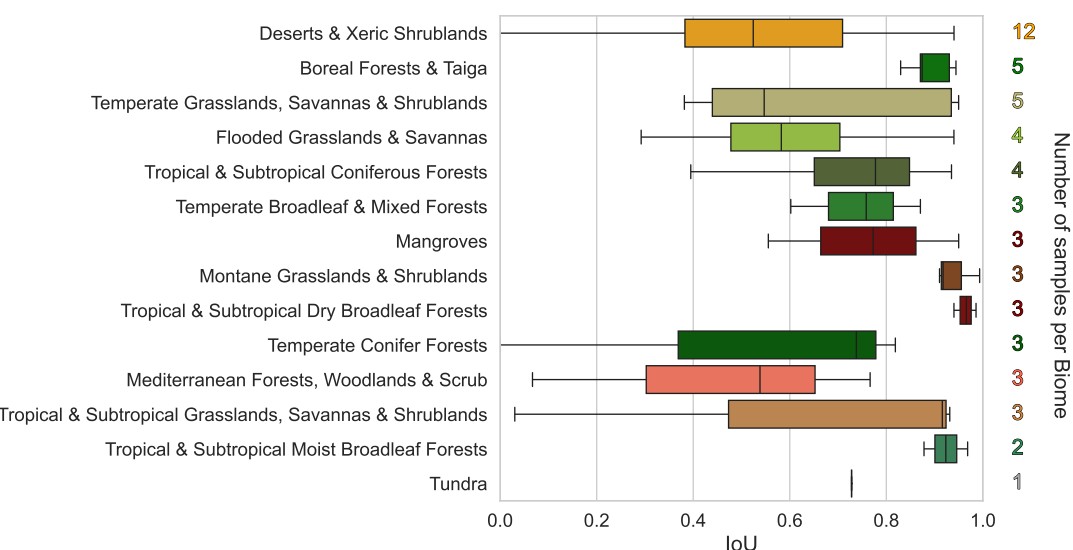

**Figure 7.** Intersection over Union (IoU) performance of the Dynamic World (DW) water class across different biomes. The number of samples per biome is shown on the right of each bar. Higher IoU scores suggest better performance in detecting surface water. The error bars represent the standard deviation of IoU scores within each biome.

Figure 9 illustrates the performance of the water class in the Dynamic World product across different biomes using IoU. IoU provides an assessment of the overlap between the predicted and ground truth water pixels, with higher values indicating better performance. The number of samples per biome varies, with some biomes having more representative data than others. For biomes with a larger number of samples, such as Deserts & Xeric Shrublands and Boreal Forests & Taiga, the IoU scores provide a more robust evaluation of the DW water class performance. Despite the variations in sample size, notable differences in performance can be observed among the biomes. It is important to note that the IoU metric is influenced by the amount of water present in each label. Higher water percentage often leads to higher IoU. However, our dataset has an average of 26.1% surface water pixels, providing a balanced assessment of the DW water class performance.

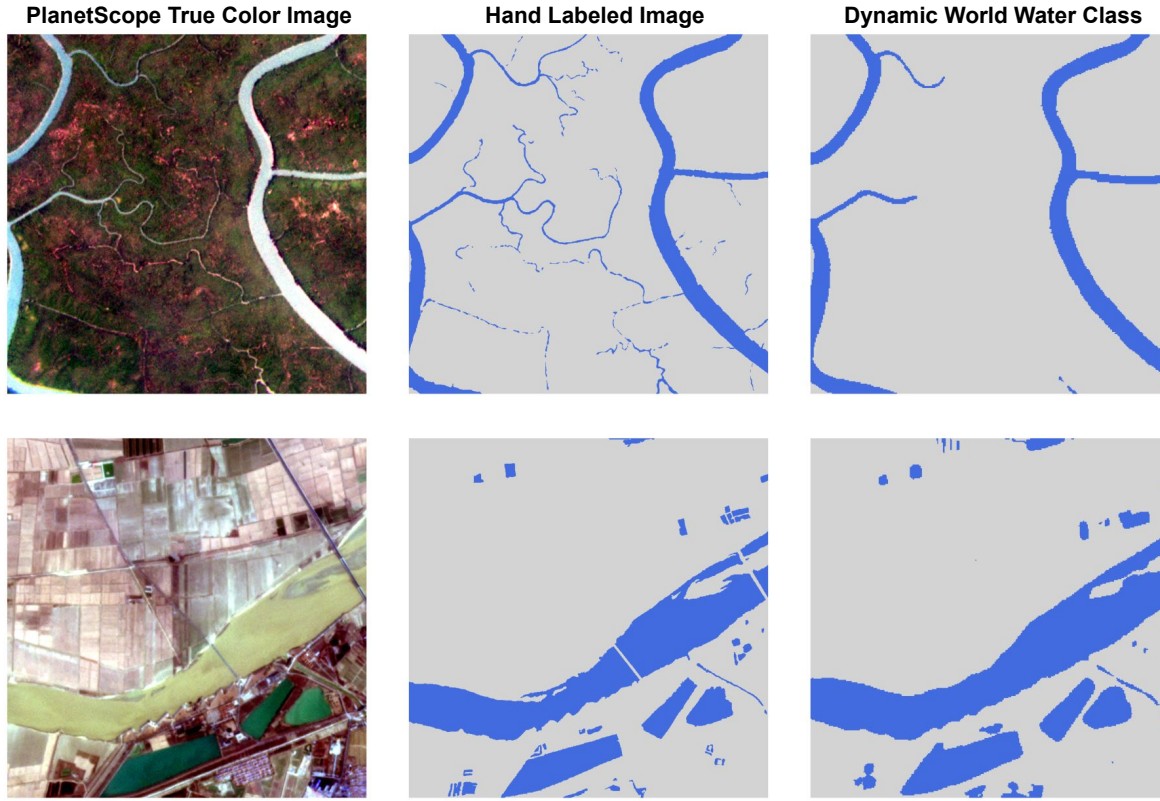

| PlanetScope True Color Image | Hand Labeled Image | Dynamic World Water Class |

**Figure 8.** Comparison of a PlanetScope true color image (left), the corresponding hand-labeled image (middle), and the Dynamic World water class prediction (right). Top: Sundarban National Park, Bangladesh (SID01), Bottom: Shandong, China (SID13).

Figure 8 provides a visual comparison of the Dynamic World water class predictions with the hand labels for two locations: Sundarban National Park, Bangladesh (SID01) and Shandong, China (SID13). The DW product appears to capture the majority of the water pixels accurately, however, it misses the narrow rivers (SID01) and the incorrectly ignores two bridges (SID13).

**Table 2.** Performance metrics for the Dynamic World (DW) water class evaluated on our hand-labeled dataset. The table presents the mean and standard deviation of various metrics. IoU denotes Intersection over Union. Higher values indicate better performance.

| Metric | Mean | Std Dev |
|:---:|:---:|:---:|
| Precision | 0.8812 | 0.2301 |
| Sensitivity | 0.7745 | 0.2830 |
| Specificity | 0.9656 | 0.0888 |
| F1 Score | 0.7970 | 0.2623 |
| IoU | 0.7216 | 0.2763 |
| Accuracy | 0.9529 | 0.0542 |

Table 2 summarizes the performance metrics for the Dynamic World water class evaluated on our hand-labeled dataset. The mean precision of 0.8812 indicates that, on average, 88.12% of the pixels predicted as water by DW are actually water in our ground truth labels. The mean sensitivity (recall) of 0.7745 suggests that DW correctly identifies 77.45% of the water pixels in our labels. The high mean specificity (0.9656) indicates that DW accurately classifies non-water pixels, with minimal misclassification as water. The F1 score, which is the harmonic mean of precision and recall, has a mean value of 0.7970, indicating a good balance between the two metrics. The mean IoU of 0.7216 signifies that, on average, there is a 72.16% overlap between the predicted and ground truth water pixels. Lastly, the mean accuracy of 0.9529 shows that DW correctly classifies 95.29% of the pixels overall, including non-water pixels. However, the high standard deviation indicates that there is a large variability in performance for almost all metrics except Specificity and Accuracy, since they take into account the non-water pixels.

## 3.2 Performance of Sentinel-1 based deep learning model

We evaluated the performance of a deep learning model (Paul and Ganju, 2021) for inundation mapping that uses S1 radar imagery. This deep learning model was the competition winner at the NASA IMPACT challenge for flood detection challenge. Unlike Dynamic World which contained a surface water class, this method focuses on flood or more specifically inundation class. Technically, our hand-labeled dataset also labels inundation although our labels did not focus on capturing flooding. Therefore, we are not directly comparing S1 IMPACT flood model against the Dynamic World water class.

We processed radiometrically corrected S1 imagery from Alaska Satellite Facility (ASF)'s data repository using the Hyp3 API. S1 imagery was searched for each label 3 days before and after the labeled date. We clipped the S1 scenes based on the labels and then we applied the trained model to these clipped S1 scenes using the trained model. We then evaluated the predictions from the deep learning model on our labels after resampling the imagery to match the resolution of the higher resolution labels using nearest neighbor interpolation. 72 S1 images were selected for this evaluation. Note that for evaluation, we converted the low-confidence water class to water. Therefore, labels were either 0 (non-water) or 1 (water).

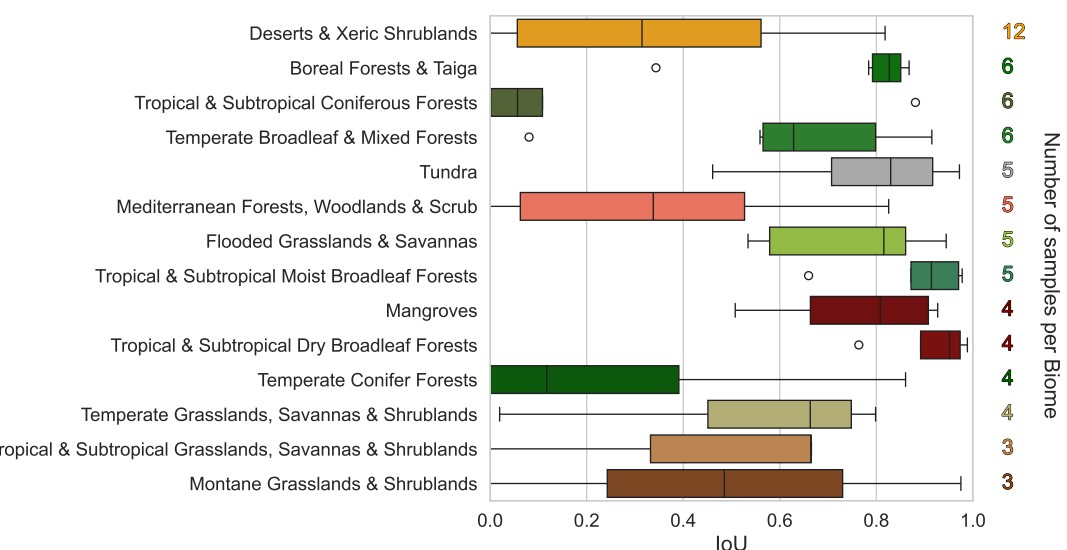

**Figure 9.** Intersection over Union (IoU) performance of the Sentinel-1 based deep learning model across different biomes. The number of samples per biome is on the right of each bar. Higher IoU scores suggest better performance in detecting surface water. The error bars represent the standard deviation of IoU scores within each biome.

Figure 9 illustrates the performance of the S1-based deep learning model across different biomes using the Intersection over Union (IoU) metric. Performance across biomes has a large variation, with some notable differences. For example, the IMPACT model performed robustly on Tropical & Subtropical Dry Broadleaf Forests, Tropical & Subtropical Moist Broadleaf Forests, Tundra, and Mangroves. Whereas for Tropical & Subtropical Coniferous Forests, Temperate Conifer Forests, and Desert & Xeric Shrublands the model performed less accurately and with large variations. Especially, Mediterranean Forests, Woodlands & Scrub where the model consistently performed poorly. The effectiveness is influenced by the fact that the training dataset of this model is focused on only 5 flood events globally. Therefore, performing accurately on the global surface water dataset is not the objective of this model. Nonetheless, the objective is still detection inundation and the variation in performance provides clues to how such a model can be improved by sampling from biomes or other contexts (urban, river, lake, etc.).

| PlanetScope True Color Image | Hand Labeled Image | Sentinel-1 IMPACT |
| :---: | :---: | :---: |

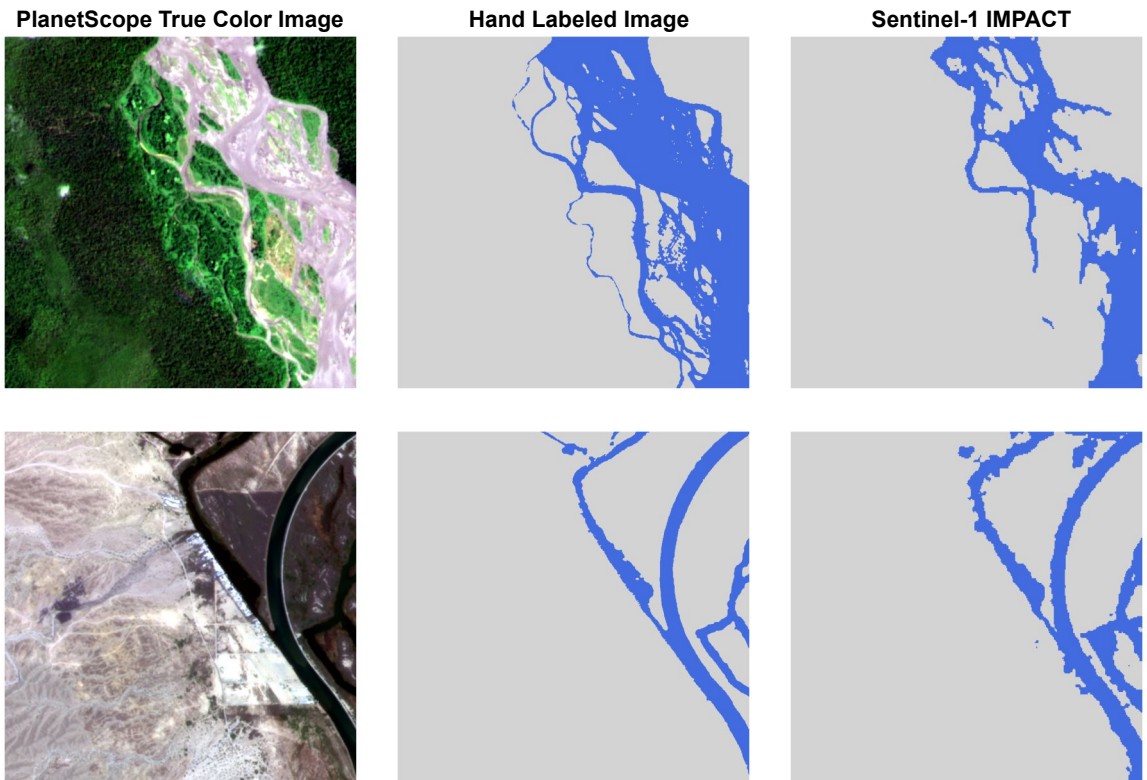

**Figure 10.** Comparison of a PlanetScope true color image (left), the corresponding hand-labeled image (middle), and the surface water predictions of the Sentinel-1 based deep learning model (right). Top: Nam Dinh, Vietnam (SID33), Bottom: Paymaster Landing, California, USA (SID59).

Figure 10 provides a visual comparison of the Sentinel-1 based deep learning model's predictions with the ground truth labels for two locations: Nam Dinh, Vietnam (SID33) and Paymaster Landing, California, USA (SID59). The model appears to capture the majority of the water pixels accurately. However, the labels and the corresponding prediction by S1-based model demonstrates the complexity of labeling and identifying water in a meandering braided river (SID33). In case of SID59, the S1 model performs well except for the coarser edges of a river in a more arid landscape.

**Table 3.** Performance metrics for the Sentinel-1 (S1) IMPACT flood detection model evaluated on our hand-labeled dataset. The table presents the mean and standard deviation of various metrics. IoU denotes Intersection over Union. Higher values indicate better performance.

| Metric | Mean | Std Dev |
|:---:|:---:|:---:|
| Precision | 0.6547 | 0.3488 |
| Sensitivity | 0.7485 | 0.3408 |
| Specificity | 0.8653 | 0.2309 |
| F1 Score | 0.6579 | 0.3435 |
| IoU | 0.5761 | 0.3406 |
| Accuracy | 0.8734 | 0.1922 |

Table 3 summarizes the performance metrics for the S1 based deep learning model evaluated on our hand-labeled dataset. The metrics exhibit significant variability across the evaluated labels. The S1 IMPACT model generally found it difficult to predict water pixels across several biomes. Apart from the differences in resolution, turbulent water and water located in spatially heterogeneous landscapes are more complicated to detect. Given the cloud free observations, S1 based models can be of considerable benefit for regular monitoring and consistent observations.

## 4 Limitations

Although our hand-labeled dataset provides a valuable resource for evaluating surface water extent products, it has several limitations that must be considered. First, the spatial resolution of the dataset is limited to 3m, making it more suitable for evaluating lower spatial resolution imagery (> 3m). For higher resolutions (<= 3m), the influence of human labeling errors on the evaluation results is likely to increase. Despite our efforts to cross-reference multiple sources of higher resolution (<1m Bing and Google basemaps) during our labeling process and implement considerable quality control, the dataset unavoidably contains biases from our labelers, in addition to the biases in the optical PS imagery itself. A model using PS will likely perform the best since PS was the primary source for labeling. Moreover, some features remained unresolved, especially features finer than 3m, leading to the addition of another class called "low-confidence water".

While we made an effort to include samples from diverse contexts in which water can be found (urban, lakes, braided rivers, mountainous regions) and multiple biomes covering different seasons, designing a truly representative dataset is not feasible. The stratified random sampling strategy used to create the dataset aims to cover diverse contexts and biomes but may not capture all the variability in surface water appearance across different regions and seasons. Additionally, the dataset only represents a snapshot in time and does not account for temporal changes in surface water extent, which can be significant in some regions due to seasonal variations, human interventions, or flooding. For example, this dataset does not include frozen water bodies.

Therefore, we recommend using evaluations from multiple independent datasets from various sources to achieve further robustness in evaluation. While our dataset is primarily designed for validation purposes, it can still be used for fine-tuning

pre-trained models. However, it does not include the original input PlanetScope images of our labels, which are required for training models. This ensures that there is no data leak from the training process, maintaining the integrity of the evaluation process. Nevertheless, relying on a single dataset for evaluation has its limitations, and using multiple independent datasets is crucial for assessing the robustness and generalizability of surface water mapping methods.

## 5   Discussion and Conclusions

In this study, we have presented a globally sampled, high-resolution surface water dataset consisting of 100 hand-labeled images derived from 3-meter PlanetScope imagery. Our dataset covers diverse biomes and contexts, including urban and rural areas, lakes, rivers, braided rivers, and coastal regions. The thorough labeling process, which involves cross-referencing multiple data sources and extensive quality control, ensures the reliability of the labels. These characteristics make our dataset a valuable resource for evaluating the performance and robustness of surface water mapping methods across a wide range of landscapes.

By applying our dataset to the S2-based Dynamic World and S1-based NASA IMPACT models, we demonstrated its utility in identifying the strengths and limitations of different satellite imagery products and methodologies. The variability in performance across biomes highlights the importance of using representative validation data to assess the spatial generalizability of mapping methods. Our findings underscore the need for multiple independent validation datasets to comprehensively evaluate surface water products and build trust in their results.

Accurate and reliable monitoring of surface water resources is crucial for sustainable water management, climate change adaptation, and conservation efforts. High-quality validation datasets like ours play a vital role in advancing these goals by enabling the development and assessment of more effective mapping methods. We anticipate that our dataset will contribute to improving the accuracy, robustness, and spatial generalizability of surface water mapping products, ultimately supporting better-informed decision-making and more efficient management of our precious water resources in the face of growing global challenges.

*Data availability.*  Our global surface water dataset (Mukherjee et al., 2024) used in this study is available in the CyVerse Data Commons, accessible via https://doi.org/10.25739/03nt-4f29.

*Author contributions.*  R.M., F.P., and B.T. designed the dataset, developed the sampling strategy, and structured the paper. R.M. and J.G. wrote the paper. R.W., P.S., E.A. labeled the images. R.M., R.W., P.S., E.A., and Z.Z. processed the data. Z.Z. uploaded the dataset.

*Competing interests.*  No competing interests are present.

*Acknowledgements.* This work was supported by the NASA Earth Science Division ACCESS Program [19-ACCESS19-0041]

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
