# Peer review of "A globally sampled high-resolution hand-labeled validation dataset for evaluating surface water extent maps"

_Earth System Science Data, 2023_

## Author Response (AR1)

**Revision Comments**

**Reviewer Comment #1:**

This is a well written paper describing a 3m water extent dataset that can be used for validating 10-20 m resolution water extent data sets. Unfortunately, rather than just focusing on this validation dataset, the authors also introduce a new Sentinel-1 algorithm called the Equal Percent Solution (EPS) for surface water extent delineation. My main concerns are:

> The water extent validation dataset derived from 3m PlantScope images covers 90 sites worldwide. It is certainly a valuable dataset, but still limited when applied for validating global water body data sets. For example, there are very few sites in arid and semi-arid environments where water body mapping with both optical and radar data can be very problematic. Validation datasets do not just require sites with water bodies. Also areas with no water cover should be included to arrive at representative accuracy statistics.
>
> The EPS method is essentially a threshold-based classifier for images characterized by a bi-modal distribution. This method - and its limitation – are well known. A major problem is that it does not work if there is no bi-modal distribution, i.e. it fails when in many instances, e.g., when the water extent is small or there is no water. Also, it does not work if there are many water-look-alike area.

My recommendations to improve this work are: (i) extent the validation data set to cover more diverse environmental conditions, i.e. also areas with no water bodies, (ii) rather than validating their own Sentinel-1 dataset, the authors can demonstrate the value of their validation dataset by using it to validate several published water body dataset.

The EPS method is in my view not publishable within the context of this paper. The authors should consider a dedicated paper where they work out the innovation of their algorithm compared to published algorithms in much more detail. Furthermore, the method needs much more testing and critical examination. From my knowledge of the Sentinel-1 backscatter data, I expect it to fail in many situations, e.g., when trying to apply it to area with no water or trying to map water in arid environments.

MINOR COMMENTS

Lines 46ff: Also the Copernicus Emergency Management Services offers global near-real-time flood maps based on Sentinel-1

Line 53: How does the sentence starting with Wieland et al. (2023) logically connect with the previous sentence?

Line 59: "can" instead of "could"

Line 76: Delete "of the true distribution"

Line 81: Delete "urban regions have a higher density of built-up environments" (that is obvious, isn't it?)

Line 193: Do the authors believe that muddiness is the reason for the different Senintel-1 signals? If yes, then much more evidence is necessary to support this claim.

**Response to Reviewer #1:**

Thank you for your time, careful reading of the paper, and valuable comments.

We initially wanted an unbiased surface water dataset that has a balanced dataset of all biomes. However, we agree with your suggestion that researchers may find more benefit in being able to evaluate their methods in typically less accurate environments such as arid or semi-arid regions. Therefore, we are adding 10 more locations covering semi-arid and arid environments across continents.

We agree that methods should also be evaluated on locations with non-water bodies in addition to the main category - water. However, we believe that most of our labels have (significantly) more non-water surface features than water surface features, such that this need is well covered. Moreover, since each image is 3 sq km, there might be sufficient non-water features adequately distant from surface water features for the model to be tested for false positive errors. That said, adding non-water labels is trivial with close to zero effort (no labeling required), we simply have to locate a few dry 3 sq km regions and add empty labels to the dataset. So kindly let us know your thoughts on that.

Finally, we agree that threshold-based solutions have their limitations, particularly when there are no bi-modal distributions. We also experienced similar issues with our threshold-based method. Therefore, we have decided to remove the Sentinel-1-based EPS method from this paper to focus primarily on the dataset. Instead, we can evaluate an optical product, for example, Dynamic World based on Sentinel-2 data in addition to the radar-based (Sentinel-1) GloFAS Global Flood Monitoring.

Thank you for the minor comments.

*Changes made based on Reviewer #1:*

We increased the number of labels from 90 to 100, with the Desert and Xeric

Shrubland biome now having 16 labels. We also added the rationale behind the addition. This increased the total surface water area labeled from 204 sq km to close to 250 sq km, leading us to redo the analysis for all the labels, including plots and graphs.

Since we decided to add 10 more labels, we added labels from 2023, increasing our temporal range to 2021-2023 from 2021-2022.

We removed our Equal Percentage Solution method, as suggested by our reviewer. Instead, we introduce two popular surface water detection methods - Sentinel-2 based Dynamic World Land Use and Land Cover product by Google and Sentinel-1 based deep learning method from the NASA IMPACT flood detection challenge. We also removed Sentinel-1 based GloFAS Global Flood Monitoring (GFM) from our evaluation since a significant amount of the labels were not covered by the product spatially or temporally. The Sentinel-1 based NASA IMPACT Challenge winner was its replacement. Now we have a balance of one optical-based and one radar-based method in our evaluation.

**Reviewer Comment #2:**

3m hand-labeled water product should be useful as most existing ones are >10m. However, I think authors have not fully justified their motivation in creating a new dataset, and how this intercompared with existing ones. For example, authors mentioned they used Wieland et al. (2023) has a 1m product. How does your product compare with theirs? Why bother having a new coarser resolution ones? Authors mentioned "multiple independent evaluations are needed", but does your product adds another layer of subjectivity? Or do you have other justifications to strengthen this part?

How is the temporal information in the Planet Images influence the representativeness of the labeling results? Can the authors expand on this, provide cautions to users and also maybe write down such information with the data to better inform users?

Authors claim they used a novel EPS method. However, it seems the details of the EPS method were not given. How is it accomplished? Why is it novel?

Given the above consideration, I'd suggest major revisions.

**Response to Reviewer #2:**

Thank you for your time, careful reading of the paper, and valuable comments.

Yes, we agree that our 3m validation product would be useful compared to the 10m datasets.

With regard to the advantage of our data set compared with Wieland et al. (2023) dataset, their labels were generated based on semi-automated analysis using NDWI masks, whereas ours were manually labeled by trained analysts. Therefore, our hand-labels have gone through more careful quality control checks, improving their accuracy over semi-automated labels. Their dataset is more suited to be used as weak labels for training deep learning models on surface water, whereas our validation dataset can better assess model accuracy due to the higher precision of hand-labels. Further, their dataset is not directly publicly accessible, whereas our efforts through the intention of hopefully publishing this paper are for its ease of access.

We advocate for more independent evaluation datasets, especially with different sampling strategies, even if some amount of subjectivity is introduced. Using multiple validated sources for evaluation could reduce unintended biases that may be present when assessing models on labels derived from the same satellite sensors used for training. Overall, evaluations using disparate sources (in our case, optical PlanetScope SuperDove sensors) lead to more robust algorithms by reducing sensor-specific overfitting.

Since Planet's constellation of sensors captures daily imagery, there is a higher chance of temporally co-incident acquisitions than from other sensors such as Sentinel-2, Sentinel-1, Landsat, etc. Therefore, we believe using Planet data as the source imagery for the labels adds to the advantages of this dataset.

We did provide information on the EPS method in section 3.1. However, based on the recommendations from Reviewer #1, we will remove it to focus primarily on the validation dataset.

*Changes made based on Reviewer #2:*

We added the imagery date for each of the labels in their filename to provide temporal information on the samples.

**Changes as a result of direct communication by the external reviewers (non-ESSD):**

There were some issues with the Countries tag on STAC, which are fixing.

We received permission from Planet to upload a degraded version of the original Planet data. To degrade the original PlanetScope data, we are converting higher precision surface reflectance values to uint8 data type. This reduces the possible range of surface reflectance values to only 256 possible numbers. The spatial resolution remains at 3m. Therefore, we are adding the uint8 version of the original Planet image with each of our labels.

***Changes made based on External Comments:***

We added Elise Arelleano-Thompson as a co-author since she labeled and reviewed the 10 new labels within the semi-arid regions, as suggested by our reviewer.

We performed additional quality control measures while fixing a few erroneous labels.

We performed a more comprehensive evaluation of the two methods mentioned above where we evaluated their performance across each biome. Compared to the previous manuscript where we compared using only 20 of our labels, we were able to evaluate using 53 labels (from Dynamic World) and 73 labels (from the Sentinel-1 based NASA IMPACT model).

We updated the limitations section and the Discussion and Conclusion sections based on the changes mentioned above.

We added a permanent DOI to our dataset.